# High Critical Current Density of Nanostructured MgB_2_ Bulk Superconductor Densified by Spark Plasma Sintering

**DOI:** 10.3390/nano12152583

**Published:** 2022-07-27

**Authors:** Yiteng Xing, Pierre Bernstein, Muralidhar Miryala, Jacques G. Noudem

**Affiliations:** 1Normandie University, ENSICAEN, UNICAEN, CNRS, CRISMAT, 14000 Caen, France; pierre.bernstein@ensicaen.fr (P.B.); jacques.noudem@ensicaen.fr (J.G.N.); 2Materials for Energy and Environmental Laboratory, Superconducting Materials Group, Shibaura Institute of Technology, 3-7-5 Toyosu, Koto-ku, Tokyo 135-8548, Japan; miryala1@shibaura-it.ac.jp

**Keywords:** spark plasma sintering, superconductivity, MgB_2_, nano grains, critical current density

## Abstract

In situ MgB_2_ superconducting samples were prepared by using the spark plasma sintering method. The density of the obtained bulks was up to 95% of the theoretical value predicted for the material. The structural and microstructural characterizations of the samples were investigated using X-ray diffraction and SEM and correlated to their superconducting properties, in particular their critical current densities, J_c_, which was measured at 20 K. Extremely high critical current densities of up to 6.75 × 10^5^ A/cm^2^ in the self-field and above 10^4^ A/cm^2^ at 4 T were measured at 20 K, indicating that vortex pinning is very strong. This property is mainly attributed to the sample density and MgB_2_ nanograins in connection to the presence of MgO precipitates and areas rich in boron.

## 1. Introduction

MgB_2_ is an intermetallic superconductor [1]. The superconductivity of this material was reported for the first time in 2001 [2]. This compound has excellent superconducting properties, especially its high critical current density [3,4] and trapped magnetic field [5], due to a strong pinning of the vortices. The voids, disorders and impurities present in the material can behave as pinning centers that hamper the vortex motion and minimize energy dissipation. Unlike cuprate superconductors [6,7], the J_c_ values of MgB_2_ can be strongly enhanced by increasing the number of grain boundaries while decreasing the grains’ size [8,9]. Although its transition temperature is relatively low for many applications (39 K), the processing conditions of MgB_2_-based materials are simpler and much cheaper than those of the cuprate superconductors. Otherwise, for applications, costs can be saved by using cryo-coolers apparatus above 4 K or developing technologies that use liquid hydrogen as a cooling fluid [10]. To prepare highly dense MgB_2_ bulk superconductors, spark plasma sintering (SPS) is a very interesting technique [11,12]. It is a rapid consolidation method that results in a good understanding and control of the sintering kinetics, as reported elsewhere [11]. The heat source is not external but is an electric current (AC, DC or pulsed) that flows across the die containing the powder to sinter. Simultaneously, a uni-axial pressure is applied. The main difference between SPS and conventional or hot-pressing methods is that SPS allows for the preparation of highly dense samples with grain growth control and a decreased processing time [13,14,15,16]. Concerning MgB_2_, the samples processed by SPS additionally show excellent mechanical properties [17].

In this contribution, we report the remarkable superconducting properties of MgB_2_ samples containing nanoparticles densified by SPS, especially their very high critical current density, J_c_.

## 2. Experimental Section

Two samples, namely sample A and sample R, were fabricated using the same sintering conditions. The aim was to investigate the sample reproducibility. The starting powders used to prepare the samples were Mg metal (99.8%, 325 mesh, Alfa Aesar, Haverhillm, MA, USA) and amorphous B powder (99%, <400 nm, PAVEZYUM advanced Chemicals, Dilovası, Turkey), mixed at the molar ratio of 1:2. A total of 1.5 g of starting powder was weighed and loaded into a tungsten carbide (WC) mold (20 mm diameter). Graphite foils were wrapped alongside the inner wall of the mold and inserted between the powder and two punches to facilitate demolding. The mold can be reused if employed within some temperature and pressure limits, usually under 850 °C and 500 MPa. The samples were sintered by spark plasma sintering (FCT Systeme GmbH, HD25, Rauenstein, Thuringia, Germany) in DC mode under dynamic vacuum (10^−3^ bar) with the following protocol:(i)500 °C + 260 MPa/15 min (Powder compaction);(ii)650 °C + 280 MPa/20 min (Synthesis);(iii)750 °C + 300 MPa/30 min (Sintering/Densification).

The introduction of these different steps aimed at avoiding magnesium evaporation. The heating rate was 100 °C/min, and the cooling time was about 20 min. The total processing duration was approximately 100 min. With respect to conventional graphite molds [11,12], WC molds sustain a higher pressure, allowing for a lower processing temperature and better control of the grain size. The bulks were polished before characterization. The density of the samples was measured using the Archimedes method with ethanol [18]. The crystalline phases present in the samples were analyzed by X-ray diffraction (XRD) with a Philips θ–2θ diffractometer and the monochromatic Cu-Kα radiation. The microstructure was analyzed with a scanning electron microscope (SEM, JEOL 7200, Tokyo, Japan). Small specimens with dimensions of 1.51 × 1.54 × 1.62 and 1.48 × 1.45 × 1.84 mm^3^ were cut from the two fabricated, bulks A and R, to investigate their superconducting properties with a SQUID magnetometer (Quantum Design, model MPMS, San Diego, CA, USA). The magnetic moment was firstly measured from 20 K to 40 K with an applied field of 20 Oe in order to determine the critical temperature; then, they were measured at various temperatures as a function of the applied field. The critical current density, J_c_ was deduced from the resulting hysteresis loops using the extended Bean model equation [19] for samples with rectangular sections: J_c_ = 20∆M/[a^2^c (b − a/3)]
where a and b (a ≤ b) are the width and length of the specimens, and c is their thickness. ∆M is the difference in magnetization between increasing and decreasing magnetic fields. A small parallelepiped sample with dimensions of 1.47 × 1.67 × 9.8 mm^3^ was cut from sample A; then, the electrical resistivity was measured using a DC 4-probe method with a PPMS (Quantum Design, San Diego, CA, USA) system. The contacts on the samples were made using silver paste. The current applied to the bar sample was 5 mA. The resistance was measured as a function of the temperature (10–300 K, sweep mode) while applying a magnetic field of up to 14 T.

## 3. Results and Discussion

The samples A and R showed the same properties. Their density was 2.46 g/cm^3^, corresponding to 95% of the theoretical value. The XRD of sample A reported in Figure 1 shows that the major peaks were due to the MgB_2_ phase with traces of MgO (3 wt.%, evaluated by the Maud refinement, [20]). The presence of MgO could be related either to the oxidation of Mg during the sintering process or to a redox reaction between B_2_O_3_ and Mg [21]. The B_2_O_3_ phase was already present in the boron powder and could react with Mg due to its low melting point (≈450 °C). Small peaks of residual carbon due to the residual carbon foil were also observed [16]. The MgB_2_ crystallite size as estimated from Rietveld refinement (isotropic size–strain model of Maud [22]) was 58 ± 0.6 nm, supposing a Gaussian distribution. 

The microstructure of the fracture surface of sample A is shown in Figure 2. Its chemical compositions were examined using a back-scattered electron detector (see Figure 2A), and the topographic view of MgB_2_ grains was given by the secondary electron mode (see Figure 2B). Unreacted boron and MgO zones, which might have a flux-pinning-enhancing effect, are visible in Figure 2A. We underscore the presence of compact MgB_2_ nanograins and the high number of grain boundaries seen in Figure 2B. 

The magnetic moment of sample A as a function of the temperature is plotted in Figure 3A. The onset of the superconducting transition occurred at 38.25 K, and there was a narrow transition width (Δ*T* around 1 K). As shown in Figure 3B, for low values of field B, the critical current density J_c_ showed an invariant behavior with respect to the applied magnetic field; this plateau became smaller and smaller as the temperature increased. This behavior may indicate that the grain connectivity, or the packing ratio, which influences J_c_ in the self-field [23], is very good in the samples. This behavior is well-described by the ‘collective pinning’ model [24] with a dynamic state of the ‘single vortex’ flux pinning regime. This flux dynamic regime reveals the importance of defects in nanometric dimensions at the grain boundaries in this applied magnetic field range (Figure 2B). For larger values of the magnetic field, a crossover field, B_sb_, from a ‘single vortex’ to a ‘small-flux bundle’, can be identified from the change in slope in Figure 3B. The inset in Figure 3B shows B_sb_ (T), which, as expected, is a decreasing function of the temperature. This quantity corresponds to the field for which J_c_(B) = 0.95 ∗ J_c_(0) [25]. This behavior due to the ‘boundle flux pinning’ regime for higher applied B values shows a critical current exponential decrease [23]. It is probable that in this magnetic region, the grains boundaries of the very small grains present in the material could dominate the pinning behavior in a high magnetic field [26]. At 20 K, a high J_c_ = 6.75 × 10^5^ A/cm^2^ in the self-field was measured. Remarkably, under 4 T, the critical current density was above 10^4^ A/cm^2^, which is an extremely high critical current density at this temperature. Another possibility is that the pinning was due to the tensile stresses between the MgO precipitates or the areas rich in boron visible in Figure 2A on the one hand and the MgB_2_ zones on the other hand. This phenomenon was previously observed in a SiC-MgB_2_ bulk [4], where for B ≈ 4 T and T = 20 K, J_c_ was reported to be in the range of the values presented here. 

Table 1 summarizes the best J_c_ of non-doped MgB_2_ bulks prepared by other authors, correlated with their density and grain size. As compared to these results, except for the samples prepared by high-pressure processing (HP) [27,28], a significant improvement in the critical current density in both the low and high field can be observed. This can be attributed to the strong densification of the nanograins, suggesting that it is possible to enhance the grain connectivity by increasing the pressure and decreasing the temperature applied during spark plasma sintering. In addition, the processing time is generally shorter than that required by the other techniques. The high J_c_ values recorded by Prikhna et al. (using HP processing) [27,28] are slightly higher than our results because their samples had a smaller grain size and higher density. However, the required sintering pressure and temperature were far greater than those required for the SPS process.

The same critical current densities were measured for both samples, as can be seen in Figure 3B. In addition, at 10 K, J_c_ for sample A was equal to 9.1 × 10^5^ A/cm^2^ and 1.8 × 10^4^ A/cm^2^ at 0 and 6 T, respectively. From the application point of view, we underscore J_c_ values equal to 4.6 × 10^5^ A/cm^2^ and 2.7 × 10^5^ A/cm^2^ at 25 K and 30 K in the self-field, respectively. These high J_c_ values are compatible with the use of liquid hydrogen or neon as cooling fluids cheaper than liquid helium. 

Concerning the electrical properties, the resistivity ρ as a function of the temperature is plotted in Figure 4A. It was equal to 43.8 µΩcm at 300 K and decreased to 17.1 µΩcm at 40 K. The residual resistivity ratio (RRR) was 2.56, which is close to the results obtained with other polycrystalline MgB_2_ bulks [23,34]. The upper critical magnetic field, B_c2_, and the irreversibility magnetic field, B_irr_, were determined from 90% and 10% of ρ_40K_, as reported elsewhere [34,35]. From measurements of the resistivity carried out from 6 K to 40 K at fields ranging between 0 T and 14 T, their temperature dependence could be determined and is plotted in Figure 4B. The linear extrapolation of B_c2_ and B_irr_ at 0 K was 21.3 and 18.2 T, respectively. These values are higher than those reported for undoped MgB_2_ bulks [36]. 

## 4. Conclusions

We investigated the superconducting properties of a dense MgB_2_ sample processed by reactive sintering at 750 °C under 300 MPa using unconventional spark plasma sintering—SPS. The high packing factor of 95% enhanced the grain connectivity and sample density. A remarkable increase in critical current density values was observed. This indicates a strong vortex pinning effect attributed to the formation of nanograins and a large quantity of grain boundaries. The performances could probably be improved by optimizing the applied pressure, the sintering time and the precursor doping. This will be the objectives of our future studies.

## Figures and Tables

**Figure 1 nanomaterials-12-02583-f001:**
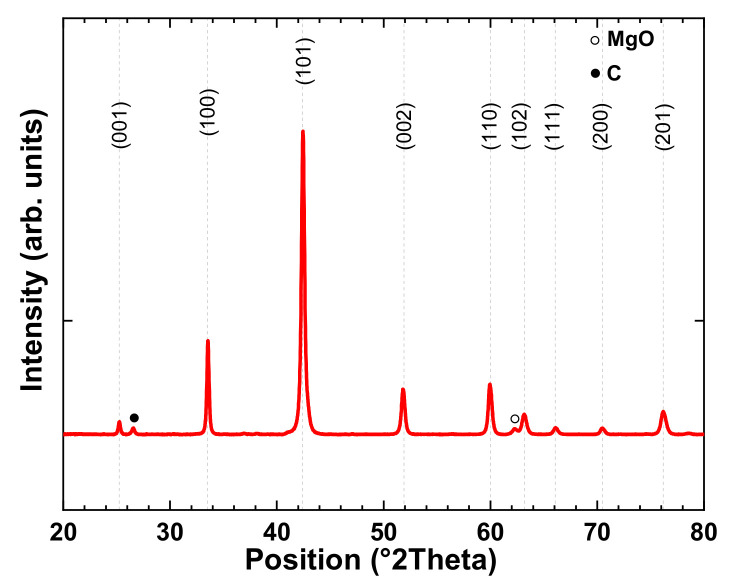
X-ray diagram of sample A.

**Figure 2 nanomaterials-12-02583-f002:**
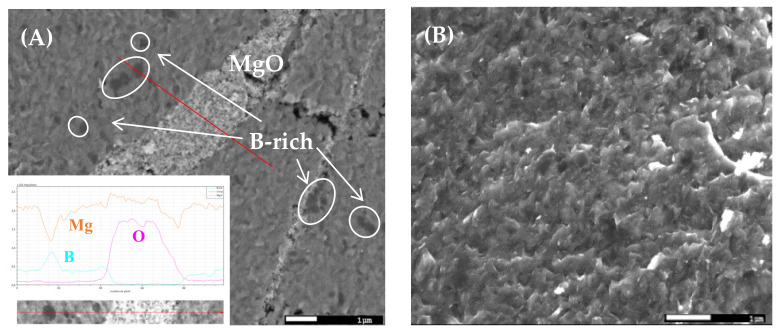
(**A**) SEM micrograph of sample A showing the unreacted boron and MgO regions in the MgB_2_ matrix. (**B**) High-magnification SEM image of nano MgB_2_ grains. The inset of Figure 2A shows the EDS red-line region scan curves of boron (cyan line), oxygen (magenta line) and magnesium (orange line).

**Figure 3 nanomaterials-12-02583-f003:**
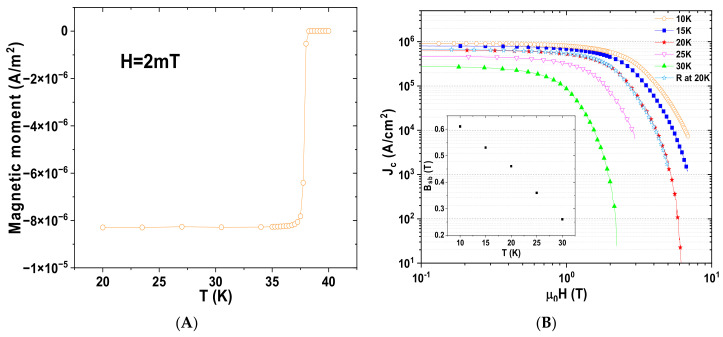
(**A**) Magnetic moment of sample A as a function of the temperature. (**B**) Magnetic field dependence of the critical current density J_c_ of sample A at various temperatures. The figure also shows J_c_ for sample R at 20 K (blue stars). The inset shows the crossover field B_sb_ as a function of temperature.

**Figure 4 nanomaterials-12-02583-f004:**
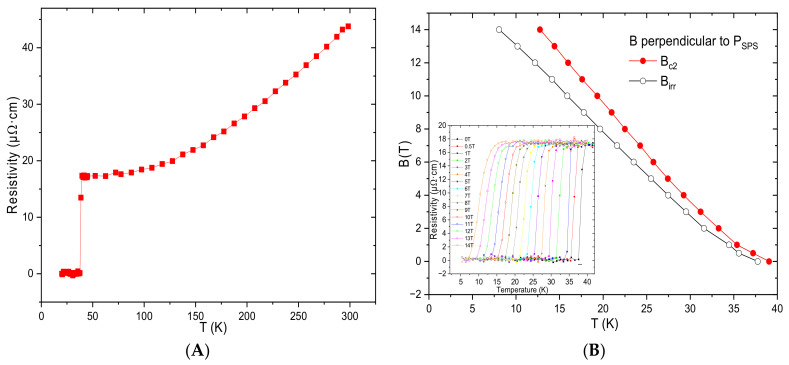
(**A**) Temperature dependence of the resistivity of sample A at 0 T with a 5 mA direct current. (**B**) Temperature dependence of B_c2_ and B_irr_ for the same sample. The inset shows the resistivity transition at fields ranging between 0 T and 14 T with a direction perpendicular to the SPS sintering pressure.

**Table 1 nanomaterials-12-02583-t001:** Characterization of MgB_2_ bulks prepared using different techniques.

Method	Sintering/ Densification	Density	Grain Size (nm)	J_c-0T_ (A/cm^2^)	J_c-4T_ (A/cm^2^)	Reference
Conventional sintering (CC)	775–800 °C 3 h	50–60%	~100	2–3 × 10^5^	10^2^–10^3^	[29,30]
Infiltration and growth process (IG)	750 °C 2 h	~90%	500	2 × 10^5^	10^2^	[31,32]
Hot-pressing or high-pressure processing (HP)	1050 °C 2 GPa 1 h	99%	17–37	5–9 × 10^5^	1–2 × 10^4^	[27,28]
Spark plasma sintering (SPS)	750–850 °C 50 MPa 20 min	75–88%	~132	5 × 10^5^	4 × 10^3^	[17,33]
In this work	750 °C 300 MPa 30 min	95%	~58	7 × 10^5^	10^4^	-

## Data Availability

Data available in a publicly accessible repository.

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
