# Peer review of "High Critical Current Density of Nanostructured MgB_2_ Bulk Superconductor Densified by Spark Plasma Sintering"

_nanomaterials, 2022, doi:10.3390/nano12152583_

Round 1
Reviewer 1 Report
This paper presents a method for producing high quality MgB2 bulk samples. The results are interesting and worth publishing in Nanomaterials.
One minor point:
In Fig 2b the scale seems wrong. Please che k
Reviewer 2 Report
The manuscript by Y. Xing et al. reports on the achieved high critical current of MgB2 bulk superconductor.
The topic of the manuscript is interesting but changes are suggests.
The abstract does not well summarized the paper, it could be improved including sizes of the investigated as well as the made characterization. I mean could be improved from many sides.
In the introduction, the authors open up a comparison with cuprates without to cite any papers. In order to make more readable the paper the authors could take in account at least the following papers: Phys. Rev. B 95, 184505 (2017), Crystallography Reports 56(1):152-156, Crystallography Reports, 2022, Vol. 67, No. 3, pp. 436–440.
The authors report even in the title the word “nanostructured” but it the paper it seems that no nanostructures are made. Please could the authors explain more about the meaning of that word in this paper? It only seems that they made a thin film but even it is not quite clean how much thick the film is.
In Fig.2 they underline regions made by different materials. It seems that the analysis is curried out by SEM tool and only looking a change of the gray scale. Since even other places in that image have a different color how can the authors be sure the intercepted zone have a phase more B.reached? I strongly believe that this analysis is misleading and it should be made at least by an EDX tool by comparing the results of different parts of the sample.
Fig.3-B reports the Jc vs. H curves at several T and it seems that the plot even includes a curve of another sample B-20K. Since it could confuse the reader, please, could the author change the name of find a better way to distinguish it?
Fig4. Shows the R-T of the made films. Could the authors explain how the temperature is changed and the current recorded and where the thermometer is placed? Did they take in account the thermalization issues before to record the data?
Archimedes method with ethanol (please add a reference).
Reviewer 3 Report
The authors in this paper show a characterization of bulk MgB2 samples realized with the Spark plasma sintering (SPS) technique. They emphasize that this technique has produced samples with very high critical current values for applied magnetic fields. They characterized the composition, grain size, different precipitated present in the samples using X diffraction and SEM techniques. Moreover, shows a very high critical temperature of Tc = 38.25K.
The trends of the critical current as a function of the applied DC magnetic field and temperature show high current density values for magnetic values at medium amplitude in particular, underline for B=4T an amplitude of Jc≈1E4A/cm2. From resistive measurements in the DC field as a function of temperature they deduce the Bc2 and Birr field trends, extrapolate their value to T=0K and affirm that these are also the highest ever produced.
The most important result of the paper is the high value of Jc for a magnetic field of 4 Tesla, however the paper has need of some general review references on the MgB2 superconductor and a more in-depth comparison between the data presented and the data published with similar measurements.
The paper is presented in a confusing and approximate way.
In my opinion, the critical current measurement described in the abstract is not clearly highlighted in the text.
In this form it is not publishable.
They need to introduce some clarifications, corrections and at least new suggested references must be introduced.
Several changes are required and mandatory.
I suggest some profound reviews and give some guidance on data analysis. I hope that the understanding of the suggested variations will not be too complex for the authors, but the paper must be revised to be published.
Below is a detailed list of the corrections that must be introduced line by line, they concern both simple corrections and substantial changes.
LEGEND:
- text of the authors in italic black color, dimension: size 12
- Variation suggested to be introduced in red color, dimension: size 12
- References to be included in red color, dimension: size 10
- My comments and indications in blu color, dimension: size 10
ABSTRACT:
Remove the last statement because, I believe is erroneous, larger defects such as MgO precipitates and areas rich in Boron should be effective at 4T tesla, as I suggest below
1. Introduction
1) In the initial line after the first sentence: ‘MgB2 is an intermetallic superconductor’ insert one or more general review references on the MgB2 compound, for example:
[1] Cristina Buzea and Tsutomu Yamashita, ‘Review of the superconducting properties of MgB2’, Supercond. Sci. Technol. 14 (2001) R115–R146
2) then continue by inserting here the sentence of the date of the MgB2 discovery introduced by the authors a few lines later: ’ The superconductivity of this material was reported for the first time in 2001 [ 4->2]’
3) insert the sentence: ‘and excellent superconducting properties, especially high critical current density [2->3, +4] and trapped magnetic field [3->5]’
(in fact here, it is necessary to introduce a further review reference where high critical currents comparable with those that will be presented in this paper): [4] Wenxian Li and >Shi-Xue Dou, ‘High Critical Current Density MgB2’ Chapter 6 in book’ Superconductors’ edit by Alexander Gabovich, (2015), http://dx.doi.org/10.5772/59492
Now in the text also specify because in MgB2 is present an 'Trapped magnetic field': this is due to the presence in MgB2 of a significant 'flux pinning'
4) Introduce in this point, if possible, some notes on flux pinning/defects meaning
5) Authors must enter a reference when they cite this cryogenic technique: ‘…or developing technologies that use liquid hydrogen as cooling fluid [6?].’
6) check and change ref numbers [5->7,6->8]
7) The sentence in the first line: 'which has good mechanical properties [1->9] must be moved here, when the SPS technique is introduced by the authors since the reference [1] used by authors refers to the mechanical properties achieved in MgB2 using the SPS method
8) check and change ref numbers [7-10->10-13]
2. EXPERIMENTAL
9) separate the two words in the sentence: ‘…..wrapped along side….’
10) change the sentence: ‘using the extended Bean model equation [11] for rectangular samples:’, with the sentence: ‘using the extended Bean model equation[11->14] for samples with rectangular section’
11) In the sentence ‘Small rectangle sample’ replace the word ‘rectangle’ con ‘parallelepiped’
12) When you talk about the ‘electrical resistivity was measured by a DC 4-probe method’ specify if the 4 contacts on the MgB2 sample had been evaporated with paid Al, Au and solder the contact with Sn, or using 'silver print' paint or other. Specify better the measurement technique, such as the current value used, etc.
RESULT AND DISCUSSION
13) check and change ref numbers (12->15, 13->16, 10->13)
14) ’The MgB2 grain size as estimated from refinement was 58 nm.’ The authors need to specify and clarify how they got this value. Have you used Scherrer's method? What is the error on the grain size? The value of 58nm will be an average value, how is the distribution?
15) Second line pag.5: ‘there is a narrow transition width’ enter the DT value deduced from the plot during the superconducting transition deduced from the plot figure 3A.
16) Figure 3B, Jc versus B at different temperatures is the heart of the article emphasized in the title, authors must give more emphasis in the discussion of the figure. Authors should begin with a general comment describing the patterns of the figure associated with the type of flux pinning / defects.
For example I suggest introduce the following discussion with some comments in the text:
As shown in fig.3B for low values of field B, the critical current density Jc, shows an invariant behavior respect to the applied magnetic field, this plateau becomes smaller and smaller in magnetic field B with the increase in temperature. This behavior may indicate that the grain connectivity, or packing ratio which influence Jc in self-field [14->17], is very good in this type of MgB2 produced with the SPS technique. This behaviour is well described by the ‘collective pinning’ model ([18] Blatter G, Feigelman MV, Geshkenbein VB, Larkin AI, and Vinokur VM, Vortices in high-temperature superconductors, Rev. Mod. Phys. 1994; 66(4) 1125-1388.] with a dynamic state of 'single vortex' flux pinning regime. This flux dynamic regime indicates the importance in this applied magnetic field range, of the small defects at 'grain boundary' with nanometric dimensions (Figure 2B). While for larger values of magnetic field B, a Hsb value can be identified from the change in slope in figure 3B. This is a crossover from ‘single vortex’ to ‘small-flux boundle’ in the flux pinning dynamics. This behavior due to the 'boundle flux pinning' regime for higher applied B values, shows a critical current exponential decreasing [18], it is probable that in this magnetic region different grain boundaries connected to larger defects have importance, these ‘dominate the pinning behavior at high magnetic field [15 ->19]’. It can be assumed that these flux pinning are connected with the large areas of MgO of tens of microns or with other defects slightly smaller around the micron similar to those regions rich in boron as shown in figure 2A.
17) The authors must introduce a plot of this Hsb versus T value deduced from figure 3B if is possible
18) …Now I suggest continuing with the discussion of the value of Jc at 20K with a comparison of the results of Figure 4 of the indicated reference 4:
a) ‘At 20 K, a high Jc = 6.75 105 A/cm2 in self-field was measured. Remarkably, under a 4T applied field the critical current density at the same temperature was above 104 A/cm2’, this is an extremely high critical current value for temperatures of 20K and suggests a strong flux pinning where, as previously mentioned, the defects due to MgO precipitates and areas rich in Boron will introduce a tensile stress that will be effective and this type of defect will be the source of a strong flux pinning between the MgO and MgB2 zones. This happens similarly in SiC-MgB2 bulk of the ref [4], where for magnetic fields B≈4T is showed a Jc value of similar values presented in this paper under identical conditions of 4T and 20K, (see fig. 4 of the ref.4).
b) continue with the discussion of Table 1 ...
19) check and change ref numbers (16-> 20, 17->21)
20) the references 18,19,20,21,22,23 are not cited, the authors must check the references
21) check and change ref numbers (14-> 17, 24-26>?)
22) In the final statement ,‘The linear extrapolation of Bc2 and Birr at 0K is 21.3 and 18.2 T, respectively. These values are higher than those reported by some authors on MgB2 thin films and bulk samples [25,26]’.
The authors need to give less emphasis because about the bulk it could have a valid certainty, in fact see figure 19 in the initial review ref that I suggested ([1] Cristina Buzea and Tsutomu Yamashita, ‘Review of the superconducting properties of MgB2’, Supercond. Sci. Technol. 14 (2001) R115–R146). While for MgB2 films this statement is not always valid (see figure 19 in ref. [1]). Much less is valid this affirmation for the wires, in fact in reference 140 fig.5 of a second review suggested ([4] Wenxian Li and >Shi-Xue Dou, ‘High Critical Current Density MgB2’ Chapter 6 in book’ Superconductors’ edit by Alexander Gabovich, (2015), http://dx.doi.org/10.5772/59492)) if extrapolated to T = 0K, much higher values are reached in the context of SiC doping in MgB2.
Round 2
Reviewer 2 Report
Dear authors I suggest to adjust the phrase "The density of the obtained bulks was up to 95 % of the theoretical density of the material" with "The density of the obtained bulks was up to 95 % of that theoretical predicted for the material."
Author Response
Dear referee,
Thank you for your suggestions of the revised manuscript. Please find our responses to your comments below.
Point 1: Dear authors I suggest to adjust the phrase "The density of the obtained bulks was up to 95 % of the theoretical density of the material" with "The density of the obtained bulks was up to 95 % of that theoretical predicted for the material."
Based on your suggestions, the modification has been down.
We thank you again for your remarks.
Reviewer 3 Report
The form and contents of the corrections introduced in the revised paper are good and I recommend its publication
Author Response
Dear referee,
Thank you again for your kind remarks and recommendation on our paper.